# MicroRNA Regulatory Pathways in the Control of the Actin–Myosin Cytoskeleton

**DOI:** 10.3390/cells9071649

**Published:** 2020-07-09

**Authors:** Karen Uray, Evelin Major, Beata Lontay

**Affiliations:** Department of Medical Chemistry, Faculty of Medicine, University of Debrecen, 4032 Debrecen, Hungary; evelinmajor@med.unideb.hu

**Keywords:** miRNA, actin, myosin, actin–myosin complex, Rho kinase, cancer, smooth muscle, hematopoiesis, stress fiber, gene expression, cardiovascular system, striated muscle, muscle cell differentiation, therapy

## Abstract

MicroRNAs (miRNAs) are key modulators of post-transcriptional gene regulation in a plethora of processes, including actin–myosin cytoskeleton dynamics. Recent evidence points to the widespread effects of miRNAs on actin–myosin cytoskeleton dynamics, either directly on the expression of actin and myosin genes or indirectly on the diverse signaling cascades modulating cytoskeletal arrangement. Furthermore, studies from various human models indicate that miRNAs contribute to the development of various human disorders. The potentially huge impact of miRNA-based mechanisms on cytoskeletal elements is just starting to be recognized. In this review, we summarize recent knowledge about the importance of microRNA modulation of the actin–myosin cytoskeleton affecting physiological processes, including cardiovascular function, hematopoiesis, podocyte physiology, and osteogenesis.

## 1. Introduction 

Actin–myosin interactions are the primary source of force generation in mammalian cells. Actin forms a cytoskeletal network and the myosin motor proteins pull actin filaments to produce contractile force. All eukaryotic cells contain an actin–myosin network inferring contractile properties to these cells. Interactions between actin and myosin are crucial for normal functions at both the cellular level, including maintenance of cell shape, cell division, cell movement, and cellular response to external stimuli, and the tissue level, including maintenance of tissue integrity, embryonic development, regulation of barrier functions, and muscle contraction [1,2]. In fact, the actin–myosin network is important in a very wide range of functions; therefore, a wide range of diseases arise from defects in the actin–myosin cytoskeleton in both muscle and non-muscle cells. In cardiac muscle cells, defects in the actomyosin cytoskeleton lead to dilated cardiomyopathy and hypertrophic cardiomyopathy [3,4,5]. In vascular smooth muscle cells, disruption in actin–myosin interactions can cause a variety of vascular diseases, including thoracic aneurysms and dissections, coronary artery disease, and stroke [6,7,8,9]. Airway smooth muscle contractile defects contribute to asthma [10,11]. Cytoskeletal abnormalities in intestinal epithelial cells contribute to inflammation-induced gut barrier disruption [12]. Changes in the actin–myosin network in mesenchymal cells influence tumor progression, cell invasion, and metastasis [13].

In light of the wide range and number of diseases influenced by actin–myosin dynamics, understanding how the actin–myosin network is regulated is crucial. Here we focus only on miRNA regulation of the actin–myosin network. We will first give a brief overview of the actin cytoskeleton and myosin heavy chains. Then, we will focus on miRNA regulation of the actin–myosin network, emphasizing regulation in a few cell types, including muscle cells, hematopoietic cells, and malignant cells. We discuss the targets of miRNA regulation, the effects on actomyosin interactions, and the contribution of miRNA regulation on the relevant pathology.

## 2. Regulation of Actin and Myosin Cytoskeleton

### 2.1. Actin Cytoskeleton 

The actin cytoskeleton consists of actin filaments, along with actin accessory and regulatory proteins. In non-muscle cells, actin cytoskeletal plasticity is necessary for cell motility, differentiation, and division, and movement of intracellular organelles. In muscle cells, the actin cytoskeleton also plays a fundamental role in muscle contraction [14]. Actin filaments are assembled from globular actin (G-actin) into double helical actin filaments (F-actin), which are linear and polar, with a fast-growing plus end and a slow-growing minus end [15,16]. 

Actin filaments are highly plastic, and assembly and disassembly of actin filaments are highly regulated to respond to the needs of the cell. Regulatory proteins control actin filament dynamics, including nucleation, elongation, and disassembly. While spontaneous nucleation of actin filaments can occur, the process is dampened due to the sequestration or binding of actin monomers by members of the profilin family or thymosin [17]. The actin-related protein (ARP) 2/3 complex mimics actin trimer/tetramers to nucleate actin filaments [18]. Nucleation promoting factors (NPF) activate the ARP2/3 complex [19]. Formin family members can also act as nucleators, stabilizing actin dimers and recruiting profilin–actin complexes [20]. Formin family members, which bind profilin–actin complexes, also participate in elongation of actin filaments [20]. In addition, Ena/VASP family members participate in actin filament elongation [21]. Disassembly of actin filaments is facilitated by coronin, glial maturation factor (GMF), and actin depolymerizing factor (ADF)/cofilin family members [22,23]. Small GTPases also play a major role in the regulation of actin dynamics, as highlighted in a later section.

### 2.2. Myosin 

Molecular motors, such as myosin, kinesin, and dynein, transduce chemical signals into mechanical force. Myosin moves along actin filaments to generate force in a process involving hydrolysis of ATP and sequential changes in the conformation of myosin while releasing and rebinding to actin. There are almost 40 different genes that encode for myosin heavy chains, and most cells express multiple myosin genes [24]. The specific combination of different myosin isoforms in a cell influences the organization of the actin cytoskeleton and, therefore, cell shape and movement and contractile properties, in the case of smooth, cardiac, and skeletal muscle. 

Myosins have a three-domain structure: A motor domain, a neck domain, and a tail domain. While the motor and neck domains have a conserved structure, the tail domain at the C-terminal of myosin is highly divergent, enabling the many cellular functions of myosins [25]. The heterogeneity of the C-termini, sometimes described as the cargo-binding site, allows binding to a variety of molecules, including proteins and lipids. For example, myosins bind to Rab proteins for vesicular trafficking [26]. Other myosins bind to lipids to allow interaction with membranes.

All myosin heavy chains have a conserved N-terminal motor domain (also known as the catalytic head domain). The motor domain interacts with actin and the conformation of the motor domain changes with nucleotide binding. In the ADP-bound state or in the absence of nucleotide, the motor domain interacts strongly with actin. The affinity for actin decreases drastically when ATP is bound to the motor domain. In this way, the ATPase cycle regulates the association of myosin with actin filaments, and, thus, the motor function of myosins. 

Myosins are classified based on the characteristics of the motor domain. Changes in the expression of myosin heavy chain isoforms can change cellular function. Mammalian myosin isoform nomenclature is based on the recommendations of Rossi et al. [27]. Class 1 myosins (MYO1A-H) have a single tail domain, which links the plasma membrane or vesicles to actin filaments [24]. While some class 1 myosins interact through adaptor proteins, others interact with lipids to modify membrane tension [28]. The MYO1 isoforms perform diverse functions, including cell crawling, actin organization on organelles, cortical tension of the plasma membrane, and endocytosis and exocytosis, and may play a role in metastasis [13,29,30,31,32]. For example, MYO1B regulates actin interactions with post-Golgi carriers and endocytic vesicles and the development of cortical tension at the plasma membrane [29,33,34]. MYO1C regulates endocytosis and exocytosis [31,35].

Class 2 myosins, classified into muscle and non-muscle isoforms, are present in all mammalian cells. Muscle myosin 2 isoforms are responsible for skeletal, smooth, and cardiac muscle contractions and are double-headed, containing 2 heavy chains and 2 light chains. Non-muscle myosin 2 isoforms (2A-C encoded by *MYH9, MYH10,* and *MYH17*) form highly dynamic short filaments [36,37] and function in a variety of activities, including cell migration, cell–cell adhesions, cell shape determination, endocytosis, and vesicular trafficking [38]. 

Next to the motor domain of myosin is the neck domain, sometimes called the lever, which binds myosin light chains (MLC), the essential light chains, and calmodulin and calmodulin-like proteins that stabilize the α-helical structure of this region. In smooth muscle myosin and non-muscle myosins, phosphorylation of the regulatory MLC, induced by binding of calcium to calmodulin, modulates the interaction between the myosin heavy chain and MLC. Phosphorylation of MLC plays a secondary role in skeletal muscle contraction. The phosphorylation of MLC is highly regulated by many kinases and phosphatases. The level of MLC phosphorylation, which can affect the force of contraction, results from a balance between MLC kinase (MYLK) and MLC phosphatase (MLCP). MYLK is predominantly activated by Ca^2+/^calmodulin, although there is evidence of regulatory phosphorylation of MYLK [39]. MLC phosphatase is a trimer consisting of a catalytic subunit (PP1c), a myosin targeting subunit (MYPT1), and a small subunit. MYPT1 has two inhibitory phosphorylation sites, T696 and T853 (human numbering); phosphorylation at these sites inhibits phosphatase activity of the holoenzyme. Several protein kinases can phosphorylate these 2 inhibitory phosphorylation sites, including Rho-associated protein kinase (ROCK), zipper-interacting protein kinase (ZIPK), and integrin-linked kinase (ILK) [40]. An endogenous inhibitor of MLCP also exists, called CPI17. Phosphorylation of CPI17 induces binding to MLCP, resulting in inhibition of phosphatase activity. Protein kinase C (PKC), ROCK, and ILK all phosphorylate CPI17 [41,42,43]. 

### 2.3. Regulation of Actin–Myosin Dynamics by Small GTPases

Small GTPases play crucial roles in regulating actin and myosin dynamics. Small GTPases cycle between the active GTP bound form and the inactive GDP form. Guanine nucleotide exchange factors (GEFs) catalyze the exchange of GDP for GTP to facilitate activation of GTPases, and GTPase-activating proteins (GAPs) promote GTP hydrolysis to inactivate GTPases, although GTPases also have intrinsic phosphatase activity. The effects of RhoA and Rac1 on cytoskeleton rearrangements have been widely established and these small GTPases interact in many cellular functions involving changes in the cytoskeleton [44]. Downstream targets of RhoA and Rac1 are Rho-associated coiled coil kinase 1/1 (ROCK) and p21-activated kinase (PAK1), which both function in actin dynamics and myosin function [44]. For example, ROCK and PAK1 phosphorylate MYPT1 to regulate MLC phosphorylation. In addition, ROCK regulates profilin and cofilin, which are involved in actin polymerization/depolymerization [45]. 

## 3. MicroRNA Overview

MicroRNAs (miRNAs) are evolutionarily conserved small regulatory RNA molecules that modulate gene expression by diverse post-transcriptional processes. Their actions depend on sequence-specific interactions of individual miRNAs with the 3′-untranslated regions of their mRNA targets. Perfect Watson–Crick complementarity is observed in 7 consecutive base pairs of the target binding region in most cases. Therefore, a single nucleotide change in this region may cause sufficient disruption in binding to deregulate the target genes. According to the latest release of the miRBase database (v22, http://mirbase.org/cgi-bin/browse.pl), the human genome contains 1917 annotated hairpin precursors and 2654 mature sequences [46]. Alles and coworkers showed that there are 2300 true human mature miRNAs, 1115 of which are annotated in the miRBase (v22) [47]. Based on the latest GENCODE (v34, https://www.gencodegenes.org/human/stats.html) data, there are 60,669 genes in the human genome, indicating that approximately 4% of human genes encode miRNA. This estimation assumes the equal distribution of miRNAs among genes, but note that a gene can code more than one miRNA [48]. Until recently, miRNAs were believed to act solely by negative regulation of target mRNA [49]; however, increasing evidence indicates that miRNAs oscillate between repression and stimulation in response to specific cellular conditions and cofactors [50]. These exciting findings, however, have made it even more difficult to explain how miRNAs regulate gene expression. 

MiRNAs can be classified by their localization as intergenic and intragenic [51,52]. Intergenic miRNAs can be found between genes and they are transcribed mostly by RNA polymerase III [53]. On the other hand, intragenic miRNAs are embedded within exons or introns of protein-coding genes and are simultaneously expressed in the same orientation of their host genes by RNA polymerase II [54]. Moreover, a small percentage of miRNAs are found scattered among repetitive elements that are transcribed by RNA polymerase III [55]. The biogenesis of miRNAs starts with RNA polymerase II- or III-dependent transcription of a miRNA gene locus generating a long primary RNA (pri-miRNA). 

Pri-miRNAs undergo maturation processes, including 5′-7-methylguanosine capping, splicing, and 3′-polyadenylation. They have an RNA hairpin in which one of the two strands contains the mature miRNA. The hairpin is cleaved from the pri-miRNA in the nucleus by the double-strand-specific ribonuclease, Drosha, forming precursor miRNA (pre-miRNA) [56]. Pre-miRNAs are transported to the cytoplasm with the help of Exportin-5 and further cleaved by Dicer [57] into microRNA (miRNA) [58]. Dicer belongs to the RNase III class. A second cleavage in the cytoplasm produces a double-stranded RNA, where miRNA is the antisense or guide [52]. Dicer acts in cooperation with additional proteins, such as the members of the Argonaute or AGO protein family [59], a protein activator of PKR [60], and the HIV-1 TAR RNA-binding protein [61]. These proteins are necessary components in the formation of the RISC-loading complex that modulates either the specificity or the efficiency of miRNA biogenesis. MiRNAs do not function as naked RNAs; instead, they create complexes with proteins forming a ribonucleoprotein (miRNP) [62].

## 4. Direct Regulation of Actin and Myosin Gene Expression by miRNAs

MiRNAs are capable of controlling the expression levels of many cytoskeletal molecules and upstream regulatory signaling modules. MiRNAs play accentuated roles in organizing diverse aspects of biochemical pathways that govern normal cellular shape, motility, and contraction. Additionally, altered miRNA expression profiles contribute to the pathogenesis of many human diseases, including cardiovascular diseases, disorders of hematopoietic cell differentiation, and cancer development [63,64]. In this review, we will summarize the current knowledge regarding the miRNAs that directly or indirectly regulate the actin cytoskeleton and the actin–myosin complex, along with their physiological and pathological relevance (Table 1, Table 2 and Table 3).

### 4.1. Direct Regulation of Actin Gene Expression by miRNAs

In addition to downregulating gene expression, some miRNAs induce translational upregulation [65], such as in the case of the cytoplasmic β-actin gene. Although β–actin is a constitutively and ubiquitously expressed housekeeping gene, its tissue-specific expression was found in mouse neurons. Gosh et al. [66] demonstrated that the β-actin gene (*ACTB*) generates two alternative transcripts with different UTR regions by alternative polyadenylation in a tissue-specific manner. The expression level of the longer transcript is relatively low but exhibits higher translational efficiency. Moreover, it harbored a conserved mmu-miR-34/34b-5p binding site. **MiR-34** upregulates the mRNA encoding β-actin, although the modulatory interaction is complex. A sequence-specific anti-miRNA molecule or mutation in the β-actin miR-34 target site results in reduced expression, which is restored by a mutant miRNA complementary to the mutant target site [66] (Table 1). 

One common target of miRNAs is the alpha smooth muscle actin (α-SMA), the main actin isoform in smooth muscle cells. α-SMA is often used as a marker of smooth muscle cell contractile phenotype versus proliferative phenotype and increased expression of α-SMA increases the contractile potential of smooth muscle tissue. Thus, the effects of miRNA on α-SMA are probably the most widely reported in the smooth muscle actin–myosin network (Table 1). 

The intracellular domain of neuroregulin-1 (NRG-1) induces α-SMA via miR-548-f. In a complex interaction, the intracellular domain of NRG-1 recruits IKZF1 (a zinc finger protein) to the first intron of the α-SMA gene to induce circular α-SMA (circACTA) formation. CircACTA acts as a sponge for miR-548-f, relieving the repression of α-SMA expression by miR-548-f and upregulating α-SMA expression [81]. In a mouse model, miR-548-f was significantly decreased in renal arteries with intimal hyperplasia, suggesting that the dysregulation of miR-548-f may participate in intimal hyperplasia in response to vascular injury [81].

**MiR-19a** also influences α-SMA expression. In addition, miR-19a influences the expression of smooth muscle 22α (SM22α), an actin-associated protein, which helps to bundle and stabilize actin filaments [68,69]. Transformation with miR-19a mimics increased expression of α-SMA and SM22α, resulting in increased migration of vascular smooth muscle cells via targeting of RhoB [70], suggesting that miR-19a may play a role in the development of atherosclerosis.

The **miRNA, let-7g**, regulates α-SMA expression. Let-7g also regulates the expression of calponin, a calcium-binding protein that regulates actin–myosin interactions and consequently, smooth muscle contraction. Wang et al. showed that let-7g increased expression of α-SMA and calponin to maintain vascular smooth muscle contractile phenotype and, therefore, reduce the formation of atherosclerotic plaques in apoE^−/−^ mice [67]. Platelet-derived growth factor-B (PDGF-B) promotes smooth muscle proliferative, rather than a contractile phenotype. Let-7g increases α-SMA and calponin expression via downregulation of PDGF-B and mitogen-activated protein kinase kinase (MEKK1) leading to reduced interaction of KLF4 (a zinc finger protein of the Krüppel-like family) and serum response factor (SRF), which subsequently de-represses myocardin (MYOCD) and increases α-SMA and calponin expression [67].

Micro-RNA regulation of PDGF-B may also play a role in smooth muscle phenotypic switching by the drug teniposide. Teniposide increases expression of **miR-21** in vascular smooth muscle cells and partially attenuates the repressive effects of PDGF-B on SM22α and α-SMA expression [71]. Transfection of vascular smooth muscle cells with a miR-21 antagomir prevents teniposide-mediated upregulation of SM22α and α-SMA [71].

The miRNA, **miR-27a** binds directly to the α-SMA transcript in the 3′-UTR and suppresses α-SMA expression in mouse primary vascular smooth muscle cells [75]. Angiotensin II upregulates miR-27a and downregulates α-SMA expression. The knockdown of miR-27a prevents the angiotensin II-mediated switch of vascular smooth muscle cells from contractile to proliferative phenotype [75]. Interestingly, increased cyclic stretch induces secretion of miR-27a from vascular smooth muscle cells, which may target GRK6 in endothelial cells in a paracrine fashion [76] (Table 1 and Figure 1).

### 4.2. Direct Regulation of Myosin Gene Expression by miRNAs

Several miRNAs regulate skeletal muscle differentiation [83,84,85], fiber type formation [86,87], and regeneration [88] by directly regulating the expression of different myosin heavy chain (MyHC) isoforms. **MiR-1**- and **miR-133**-encoding genes form clusters that can be found on chromosomes 2 and 18 of mice. These miRNAs are absent in undifferentiated muscle cells; however, differentiation can activate their expression. MiR-1 overexpression strongly promotes myogenesis of C2C12 mouse myoblast cells by inducing the expression of MyHC, a late myogenic marker [84]. Furthermore, mutations in the seed sequences of miR-1 abolish its ability to activate myogenic gene expression. In contrast, overexpression of miR-133 inhibits MyHC expression and mutant forms of miR-133 are unable to repress the expression of MyHC [84] (Table 2 and Figure 2). 

Additionally, miR-133a is involved in a skeletal muscle fiber type switch in response to exercise training [87]. Normally, this phenomenon can be characterized by an increase in the proportion of slow-twitch type I and fast-twitch type IIA oxidative fibers at the expense of fast-twitch type IIB glycolytic fibers. Nie and coworkers demonstrated that the deletion of miR-133a caused a significant loss of type IIB fibers without a relative increase in type IIX/I and type IIA fibers in mice, resulting in impaired exercise tolerance [87]. **MiR-206** is exclusively expressed in skeletal muscle tissue and its expression is induced during differentiation. Transfection of C2C12 cells with double-stranded RNA duplexes, which can mimic miR-206 function, results in increased MyHC expression in the presence or absence of serum. Following 2′-*O*-methyl antisense oligonucleotide treatment and serum deprivation of C2C12 cells, MyHC levels are diminished due to the inhibition of miR-1 and miR-206 [83]. Both **miR-143-3p** and **miR-30e** expression are upregulated during C2C12 cell differentiation. Overexpression of miR-30e promotes C2C12 myoblast differentiation and significantly decreases the expression of myosin heavy chain type I (MyHC-I) by reducing *Pgc1α* mRNA levels and simultaneously increasing the expression of MyHC-IIA, IIX, and IIB-encoding genes [86]. In addition, miR-143-3p overexpression inhibits the expression of many differentiation markers, especially MyHC, whereas transfection of miR-143-3p inhibitors enhances myotube formation [85]. MiRNAs seem to be important regulators of skeletal muscle regeneration, even though the underlying molecular mechanisms need to be further investigated.

**MiR-501** is a muscle-specific miRNA located in an intronic region of the voltage-sensitive chloride channel 5 (*Clcn5*) gene. Western blot analysis of the embryonic myosin heavy chain (MyHC-emb) revealed a significant reduction in its protein level when miR-501 was silenced. As muscle regeneration proceeds, MyHC-emb is replaced by adult MyHC isoforms, but the latter showed lower abundance due to miR-501 inhibition [88].

Among the stress-inducible miRNAs in the adult mouse heart, **miR-208a** plays crucial regulatory roles in pathological cardiac remodeling by controlling MyHC-β expression [95,98]. MiR-208a is a cardiac-specific miRNA that is located in an intron of the MyHC-α-encoding gene. Thoracic aortic banding (TAB) causes cardiac hypertrophy by inducing MyHC-β expression and hampering MyHC-α expression. MiR-208a^−/−^ mice were incapable of upregulating MyHC-β in this model. Instead, MyHC-α expression was increased in miR-208a^−/−^ mice as a compensatory mechanism. Consistently, miR-208a overexpressing transgenic mice exhibited marked and specific upregulation of the MyHC-β-encoding gene [95]. Similarly, a significant upregulation of miR-208a in type 2 diabetic mice and the human heart was associated with an increase in MyHC-β expression [98]. MiR-208a also controls the expression of Myh7b (also referred to as MYH14) and its intronic miRNA, miR-499, in a precise, stoichiometric manner, even in the absence of stress. Myh7b/miR-499 expression was decreased by 50% in the hearts of miR-208a^+/-^ mice and was terminated in the hearts of miR-208a^−/−^ mice [99].

**MiR-195** also belongs to the stress-inducible miRNAs in the heart. Real-time PCR analysis of miR-195 overexpressing cardiac tissue showed dramatic upregulation of MyHC-β expression in transgenic mice leading to cardiac failure [93]. Additionally, forced expression of **miR-199a** in rat primary cardiomyocytes led to the downregulation of MyHC-α mRNA levels by 80%, while MyHC-β expression was not affected, indicating that miR-199a may influence cardiac function through the regulation of cardiomyocyte contractile proteins [94].

MiRNAs can also be found in melanocytes and are associated with changes in melanosome transport [92] and melanin secretion [91]. Myosin 5A (MYO5A) forms a tripartite complex together with Rab27a and melanophilin (Mlph) that is responsible for transporting melanosomes along actin filaments. Luciferase reporter assays showed that **miR-145** directly binds to MYO5A in mouse and human melanocytes. Overexpression and downregulation of miR-145 reduce and increase the expression of a set of pigmentation genes including MYO5A at the mRNA and protein levels in melan-a cells [92]. **MiR-143-5p** targets the 3′-UTR of the MYO5A transcript [91]. Upon silencing miR-143-5p with a short tandem target mimic (STTM)-miR-143-5p, MYO5A expression was significantly increased along with Mlph and Rab27a in mouse melanocytes, suggesting a stable interaction between them [91]. Dual-luciferase reporter assays in 293T cells demonstrate that miR-143-5p does not bind to the mutant form of MYO5A as a consequence of reduced complementarity [91]. Based on these data, adequate MYO5A expression is necessary for proper melanosome transport and melanin secretion.

Thyroid hormone (TH) has a fundamental role in determining skeletal muscle fiber type composition. Recently, miRNAs that are targeted by TH in striated muscles have been identified [89,90]. Zhang and coworkers demonstrated that **miR-133a1,** which controls muscle fiber type specification, is a direct target gene of TH [90]. Overexpression of miR-133a in soleus (SOL) muscles and C2C12 myotubes decreases the expression of MyHC-I with a concomitant increase in MyHC-II, while TEA domain family member 1 (TEAD1) overexpression has the opposite effect. Overexpression of TEAD1 without its 3′ UTR significantly antagonizes the effect of miR-133a mimics. By contrast, TEAD1 with its 3′ UTR shows an attenuated effect, indicating that TEAD1 mediates miR-133a effects in the regulation of muscle fiber type composition [90].

TH can also exert effects through thyroid hormone receptors (THRs). **MiR-27a** was found to upregulate MyHC-β expression but not MyHC-α expression in neonatal rat ventricular myocytes (NRVMs) via thyroid hormone receptor β1 (THRβ1) [89]. TH treatment along with THRβ1 overexpression enhances the downregulation of MyHC-β, whereas in the absence of TH or the presence of miR-27a decoys, MyHC-α expression does not change. THRβ1 overexpression also leads to the suppression of MyHC-β protein levels under serum-containing conditions, while MyHC-α levels remained unaltered. The same result was obtained using miR-27a decoys, suggesting that miR-27a can regulate MyHC-β expression by targeting THRβ1 [89].

**MiR-499** is encoded by the ancient MYH7b gene, and transcriptionally regulates the expression of its host gene via Sox6, a transcriptional repressor [97]. In the presence of miR-499, the activity of a luciferase reporter gene carrying 1 kb of the mouse Sox6 3′ UTR region was reduced by 60% compared to the control. Additionally, endogenous Sox6 mRNA levels were decreased due to miR-499 overexpression in cardiomyocytes. On the other hand, overexpression of Sox6 in C2C12 myoblasts resulted in a decrease in MYH7b expression [97]. Taken together, it can be assumed that miR-499 can control the expression of its host gene and, therefore, its expression, by targeting Sox6.

PPARβ/δ and PPARα exert opposing effects on the type I skeletal muscle fiber program through a downstream regulatory circuitry, which consists of miRNAs [96]. **MiR208b**, which is encoded by the MYH7 gene, and miR-499, which is encoded by the MYH7b gene, exert their effects on the same mRNAs to downregulate transcriptional repressors of the type I skeletal muscle fiber program, such as Sox6 [97,99]. MiR-208b/MYH7 and miR-499/MYH7b expression are markedly increased in muscle creatine kinase (MCK)-PPARβ/δ muscle but are undetectable or significantly reduced in MCK-PPARα muscle. Antisense-mediated inhibition of miR-208b or miR-499 alone does not affect MyHC expression, although inhibition of both miRNAs causes a dramatic decrease in MYH7 mRNA in wild type (WT) myotubes and blocks the enhancing effects of PPARβ/δ on slow-twitch gene expression in MCK- PPARβ/δ myotubes. Moreover, crossing MCK-miR-499 mice with an MCK-PPARα line leads to the prevention of PPARα-mediated repression of the type I muscle fiber program [96]. Collectively, this evidence suggests that PPAR signaling is upstream of miR-208b/MYH7 and miR-499/MYH7b, and that PPARβ/δ activates, whereas PPARα suppresses their expression in muscle (Table 2 and Figure 2).

Non-muscle myosin II (NMII) functions in cellular organization, motility, cell shape, and polarity, and altered NMII activity contributes to a variety of disease pathologies. Non-muscle myosin IIA (NMIIA) consists of non-muscle myosin heavy chain IIA (NMHCIIA), encoded by the MYH9 gene, and regulatory and essential light chains [100]. **Let-7f** acts as a tumor suppressor to inhibit invasion and metastasis in gastric cancers via direct targeting of the tumor metastasis-associated gene, MYH9 [101]. **MiR-6089** also targets MYH9 and its overexpression suppresses ovarian cancer cell proliferation, migration, invasion, and metastasis in vivo and in vitro, and miR-6089 negatively correlates with MYH9 expression in clinical samples. The proposed mechanism is an miR-6089/MYH9/β-catenin/c-Jun negative feedback loop [102]. Non-muscle myosin IIB (NMIIB) consists of NMHCIIB, encoded by the MYH10 gene, and regulatory and essential light chains. In migratory cells, NMIIB preferentially localizes to the rear of the cell [100]. **MiR-200a** directly targets NMHCIIB and the overexpression of NMHCIIB partially rescues miR-200a-mediated inhibition of cell migration, as well as cell growth in vitro and in vivo. Moreover, siRNA-mediated silencing of NMHCIIB expression inhibits meningioma tumor growth in mice [103]. MiR-200a also targets and downregulates the MYH10 gene product in nasopharyngeal carcinoma [104]. MyRepress, a new technology that models gene expression of miRNAs in non-conventional binding sites revealed another two miRNAs targeting MYH10: **miR-181a-5p** targets one non-conventional target site in the coding region of MYH10, while **miR-367-5p** was identified as a repressor of MYH10 expression in MCF7 cells [105]. Non-muscle myosin IIC, the most recently discovered non-muscle myosin isoform, consists of NMHIIC and is encoded by MYH14 [100]. To our knowledge, no information has been published yet concerning the miRNA regulation of this isoform or about the myosin light chain component of NM.

## 5. Physiological and Pathological Processes Related to miRNA-Regulated Pathways

The actin–myosin network is present in all eukaryotic cells and is highly plastic, facilitating alterations in response to changing cellular conditions or responses to external stimuli. Because of the wide variety of cellular activities in which the actin–myosin network participates, virtually all physiological processes are affected. A growing body of evidence demonstrates that miRNAs regulate the plasticity of the actin–myosin network under both physiological and pathological conditions. We will review only a few of these processes impacted by miRNA regulation, including smooth muscle function and vascular diseases [72,73,74,77,78,79,80,82,106,107,108,109,110,111,112,113,114,115,116,117,118,119,120,121,122,123,124,125,126,127,128,129], hematopoiesis [130,131,132,133,134,135,136,137], podocyte biology [138,139], osteoblast differentiation [140,141,142,143], cilia assembly [144,145,146,147,148], cancer formation [127,149,150,151,152,153,154,155,156,157,158,159,160,161,162,163,164,165,166,167,168,169,170,171,172,173], leukocytes and lymphocytes in action [174,175,176,177,178], and cardiovascular disorders [93,179,180,181].

### 5.1. MiRNAs in Smooth Muscle Function and Diseases

In smooth muscle cells, the actin–myosin network plays a crucial role not only in smooth muscle contractile activity, but also in smooth muscle cell motility, intracellular processes, and proliferation. Disruption of these processes can have serious pathological consequences leading to disease development. In vascular smooth muscle, the pathological consequences of disrupting actin–myosin dynamics include vascular restenosis, hypertension, and atherosclerosis [116]. Recently, a number of publications have revealed the role of miRNAs in regulating smooth muscle function via direct (discussed in the previous section) or indirect targeting of the actin–myosin network. Understanding the effects of miRNAs can lead to the identification of new drug targets for treating smooth muscle-related diseases.

In addition to the direct effects of miRNAs on smooth muscle actin and myosin (as described above), miRNAs also affect the expression of proteins that regulate the actin–myosin network in smooth muscle cells. The crucial role of miRNAs in smooth muscle function is demonstrated by Dicer knockouts. If Dicer is deleted during embryogenesis, severe hemorrhage causes 100% lethality at E16.5–E17.5 [108]. Conditional smooth muscle specific deletion of Dicer in adult mice results in decreased contractile function and a profound reduction in blood pressure [107].

Interestingly, not only do miRNAs influence actin–myosin dynamics, but actin–myosin dynamics affect miRNA levels also. In a complex feedback mechanism, actin polymerization is thought to regulate the levels of a number of miRNAs to precisely regulate contractile and cytoskeleton protein levels in response to cellular needs. Myocardin-related transcription factors (MRTFs) can shuttle between the cytoplasm and the nucleus. In the cytoplasm, MRTFs bind to G-actin, preventing nuclear translocation of these transcription factors [119,120]. Upon actin polymerization into F-actin, MRTFs are released and shuttled to the nucleus, where they interact with serum response factor (SRF) [119,120]. Alajbegovic and colleagues showed that in conjunction with MRTF, actin polymerization in human and mouse vascular smooth muscle cells regulates **miR-1, miR-22, miR-143, miR-145**, and **miR-378a** [106]. Increased aortic dilation causes actin depolymerization and reduced expression of these miRNAs [106], implying that these miRNAs may play a role in vascular diseases, such as aortic aneurysms.

Much of the effects of miRNAs on smooth muscle function are elicited by driving smooth muscle cells towards either a proliferative/synthetic phenotype or a differentiated/contractile phenotype. Dedifferentiated vascular smooth muscle cells (VSMCs) possess increased rates of proliferation and migration, as well as reduced expression of differentiation markers, such as α-SMA, SM22α, and smooth muscle myosin heavy chain (MYH11). The development of vascular diseases, such as post-angioplasty restenosis, atherosclerosis, and hypertension can be ascribed to pathological smooth muscle cell phenotype switching, which is modulated in numerous ways by miRNAs. The diverse molecular mechanisms regulating SMC differentiation by miRNAs involve the direct or indirect regulation of transcription factors, including serum response factor (SRF), myocardin (MYOCD), myocardin-related transcription factors (MRTFs), and the Krüppel-like zinc finger family (KLF). SRF, MYOCD, and MRTFs induce the differentiated contractile protein expression in smooth muscle cells. In contrast, the Krüppel-like zinc finger family (KLF) induces the proliferative smooth muscle phenotype [118] (Figure 1 and Table 3). 

**MiR-143/145,** which is located in a non-protein coding region, is widely expressed in smooth muscle cells, and, thus, has been extensively studied with respect to the smooth muscle phenotype. Genetic deletion of miR-143/145 results in a more proliferative smooth muscle cell phenotype versus a contractile phenotype, indicating that miR-143/145 favors the contractile phenotype [109]. The miR-143/145 cluster is regulated by MYOCD and SRF [77]. Knockdown of miR-143/145 results in disruption of stress fiber formation, indicating that miR-143/145 plays a crucial role in cytoskeleton assembly [77]. Among the targets for miR-143/145 that disrupt actin assembly are MRTF-B, β-actin, cofilin, KLF5, MYOCD, and ROCK1 [77,78]. Knockdown of miR-143/145 results in reduced neointimal formation and the inability to respond to vascular injury [77]. In addition, miR-145 expression is reduced in vascular smooth muscle cells from patients with atherosclerosis, supporting a role for miR-143/145 in the development of atherosclerosis [78]. The miR-143/145 cluster may also mediate an increased risk of vascular diseases induced by hyperglycemia [114]. Glucose induces the expression of smooth muscle differentiation, including calponin and SM22α, and genetic ablation of miR-143/145 ameliorated these effects [114]. In addition to the effects on vascular smooth muscle cells, miR-145 has similar anti-proliferative effects in corpus cavernosum smooth muscle cells [117].

**MiR-23b** [72] and **miR-125a-5p** [73] are highly expressed in VSMCs and are downregulated after vascular injury in vivo. Their overexpression is sufficient to reduce VSMCs proliferation and migration, and to promote the expression of selective VSMCs marker genes, such as α-SMA, MYH11, and SMA22α [72,73]. Over-expression of miR-23b decreases neointimal formation induced by a balloon injury. MiR-125a-5p is downregulated in response to PDGF-BB and targets ETS-1. These studies suggest that regulation of miR-23b and miR-125a-5p contribute to restenosis.

In vitro functional studies showed that overexpression of two miRNAs, **miR-330** and **miR-125b-5p**, have opposing actions on the renin lineage of vascular smooth muscle cells. Under normal conditions, miR-125b-5p was expressed in arteriolar SMCs and in juxtaglomerular (JG) cells, but after reacquisition of the renin phenotype, miR-125b-5p was downregulated in arteriolar SMCs. MiR-330, normally absent, is expressed in JG cells under stress and inhibits contractile characteristics of these cells, favoring their endocrine character. In silico analysis showed that miR-330 and miR-125b-5p have potential binding sites in smoothelin, calbindin 1, MYH11, α-SMA, and renin genes [80].

**MiR-182** is downregulated during the dedifferentiation of rat aortic smooth muscle cells in culture [79]. On the other hand, transfection of vascular smooth muscle cells with miR-182 increased the expression of α-SMA, SM22α, and calponin and, consequently, pushed the smooth muscle cells towards the contractile phenotype and inhibited dedifferentiation, proliferation, and migration [79]. MiR-182 targets the 3′-UTR of fibroblast growth factor 9 (FGF9) to downregulate FGF9 expression, which in turn affects the expression of the smooth muscle specific proteins [79,128]. In addition, FGF9 upregulates platelet-derived growth factor receptor B (PDGFRB), which is necessary for smooth muscle cell proliferation and migration. Thus, miR-182 may play a role in vascular proliferative diseases, such as atherosclerosis.

PDGF also pushes VSMCs towards the proliferative phenotype and promotes migration into the neointimal layer after arterial injury via **miR-663** and **miR-26a**. MiR-663 is significantly downregulated in human aortic VSMCs upon PDGF-BB-treatment, whereas miR-663 expression is markedly enhanced during VSMC differentiation. Overexpression of miR-663 results in increased expression of the molecular markers for VSMC differentiation and reduced expression of JunB and myosin light chain 9 (MYL9) [82]. **MiR-26a,** another regulator of PDGF-BB-mediated VSMC phenotypic switch is significantly increased in the PDGF-BB-stimulated VSMC model and in arteries with neointimal lesion formation. MiR-26a suppresses Smad1 expression, leading to VSMC switching to the proliferative/synthetic phenotype and eventually to vascular remodeling by downregulating VSMC differentiation marker genes [74].

During the formation of atherosclerotic plaques in mice, a parallel increase in expression levels of MYLK and **miR-92a** was observed. The activation of ROCK/STAT3 and/or MYLK/STAT3 may upregulate miR-92a expression, which subsequently inhibits KLF4 expression and promotes PDGF-BB-mediated proliferation and migration of VSMCs [186].

**MiR-128-1** is an intronic miRNA encoded for by two distinct genes, miR-128-1 and miR-128-2, which are located in the introns of R3HDM1 (R3H domain containing 1) and RCS (ARPP-21, cyclic AMP-regulated phosphoprotein, 21 kDa) [110]. MiR-128 modulates the vascular smooth muscle phenotype through targeting KLF4. KLF4 modulates the methylation of MYH11 (encoding for myosin heavy chain 11, a smooth muscle myosin), preventing its transcription [112]. Increased miR-128 expression correlates with the contractile phenotype. In a mouse model of carotid stenosis, lentiviral delivery of miR-128 suppressed intimal hyperplasia [112]. Thus, altered miR-128 expression may also play a role in vascular proliferative diseases and may be a potential target for treating these diseases.

**MiR-223** is an evolutionarily conserved miRNA located on the X chromosome [115,124]. In pulmonary artery smooth muscle cells, hypoxia downregulates miR-223, resulting in increased proliferation of the smooth muscle cells [129]. In vivo, a miR-223 antagomir attenuates hypoxia-induced increases in pulmonary artery hypertension [129]. MiR-223 targets RhoB, resulting in decreased RhoB expression, decreased ROCK activity, and, subsequently, inhibition of MYPT1 phosphorylation. In addition, miR-223 targets the 3′-UTR of MLC to inhibit MLC expression [129]. Overall, hypoxia-induced downregulation of miR-223 induces downregulation of the RhoB/ROCK pathway leading to pathological changes in pulmonary artery smooth muscle cells; thus, miR-223 may be a target to treat pulmonary artery hypertension.

Transfection of bronchial smooth muscle cells with a **miR-133a** antagomir upregulates RhoA expression [111]. Human and mouse RhoA have a putative binding site for miR-133a in the 3′-UTR; thus, miR-133a may directly regulate RhoA in bronchial smooth muscle cells [111]. Interleukin-13 downregulates miR-133a, resulting in upregulation of RhoA and, consequently, the increased bronchial smooth muscle contractility present in allergic asthma [125]. Similarly, hypercapnia also upregulates RhoA in airway smooth muscle cells, and this upregulation was prevented by transfection with a miR-133a mimic [126].

Compared to vascular smooth muscle cells, much less has been published on the role of miRNAs in organ smooth muscle phenotype and regulation of the actin–myosin network. The importance of miRNAs in gastrointestinal smooth muscle function is demonstrated by the deletion of Dicer, which causes downregulation of contractile genes in gastrointestinal smooth muscle cells [123]. Similar to vascular smooth muscle, SRF regulates differentiation of smooth muscle cells via several miRNAs in the gastrointestinal tract, including **miR-199a-3p, miR-214, miR-143**, and **miR-145** [122]. **MiR-143** and **miR-145** are upregulated in the intestinal smooth muscle of patients with gastroschisis compared to patients with atresia, and are associated with contractile dysfunction [113,121] (Figure 1 and Table 3). In cultured bladder smooth muscle cells, cytoskeleton remodeling and loss of contractility occur in conjunction with the upregulation of **miR-199a-5p [113]**. WNT2 is targeted by miR-199a-5p, which increases the expression of smoothelin and SM22 via inhibition via WNT signaling [113].

### 5.2. MiRNAs in Cardiovascular Diseases

The heart responds to both physiological triggers of increased demand, e.g., exercise, and pathological triggers, e.g., hypertension and valvular diseases, by enlarging the myocardium via cardiac muscle cell hypertrophy. Several recent publications demonstrate the role of miRNAs in the development of cardiac hypertrophy. In fact, a signature pattern of miRNAs is upregulated (miR-21, miR-23a, miR-125, miR-195, miR-199, and miR-214) or downregulated (miR-1, miR-29, miR-30, miR-133, and miR-150) during the development of cardiac hypertrophy [93]. The in vitro expression of miR-133 and miR-1 inhibits the development of cardiac hypertrophy, and miR-133 and miR-1 are downregulated in murine models and patients with myocardial hypertrophy [179]. The two major targets of miR-133 relevant to cardiac hypertrophy development are RhoA and Cdc42 [179]. Both RhoA and Cdc42 are associated with cytoskeletal and myofibrillar rearrangements during hypertrophy.

Several miRNAs can induce cardiomyocyte proliferation promoting cardiac regeneration. The most effective miRNAs activate nuclear localization of Yes-associated protein (YAP) transcriptional cofactor and induce expression of YAP-responsive genes. Besides targeting the Hippo pathway by miR-199, several pro-proliferative miRNAs inhibit filamentous actin depolymerization by targeting cofilin-2 [181]. In particular, a shift was observed between F- and G-actin toward the polymerized state in response to overexpression of miR199-3p, miR-302d, miR-373, and miR-33b. Cardiomyocytes showed rounded morphology and formed cortical layers of actin. The miRNAs acted by directly downregulating cofilin-2 [181], which activated YAP nuclear translocation and stimulated cardiomyocyte proliferation. Thus, actin cytoskeleton dynamics are a strong activator of YAP in response to mechanical cues, facilitating adaptation to the extracellular environment [180]. Several other proteins regulating the actin cytoskeleton were predicted by computational algorithms to be regulated by miR590-3p, miR-302d, miR373, miR-199a, miR33b, miR-302d, and miR-373, including twinfilin-1 and-2, thymosin, and profilin-2 [181].

### 5.3. Regulation of Actin Cytoskeleton by miRNAs in the Hematopoietic System

MiRNAs regulate hematopoiesis and the function of both myeloid and lymphoid progeny by several mechanisms. MiRNAs play a vital role in every stage of hematopoiesis. Our major focus will be those miRNAs that target actin and myosin cytoskeleton affecting the formation, development, and differentiation of blood cells, as well as their pathological relevance [137].

Numerous miRNAs participate in the process of erythropoiesis, but their major targets are the regulators of erythroid homeostasis, γ-globin gene expression, and erythroid markers [137]. Fluid shear stress is one of the major factors that promotes the maturation of erythrocytes by acting on the membrane skeleton, which requires actin cytoskeletal remodeling, creating its physiologically functioning hexagonal topology. **MiR-23-3p** is suppressed by shear stress leading to the upregulation of erythrocytes tropomodulin of 41 kDa (E-Tmod41), which caps the end of filamentous actin, and induces F-actin cytoskeleton remodeling and erythroid differentiation [136].

**MiR-142** also plays a crucial role as a broad hematopoietic pro-differentiation factor and is involved in megakaryogenesis [131], B-cell leukemogenesis [132], and dendritic cell development [135]. Megakaryopoiesis requires distinctive cytoskeletal rearrangements, including the cytoplasmic protoplatelet protrusions to bend and bifurcate, leading to the proper biosynthesis of thrombocytes and their release into the bloodstream [130]. Genetic ablation of miR-142 impairs megakaryocyte maturation, inhibits polyploidization, and induces abnormal proplatelet formation and thrombocytopenia [131]. Based on the study of Hornstein et al., actin cytoskeletal dynamics are disturbed, and more homogenous and circular F-actin structures are formed in miR-142-depleted cells relative to the control. Pivotal actin cytoskeleton-associated proteins were identified as direct targets of miR-142, including twinfilin-1, cofilin-2, integrin alpha V, glucocorticoid receptor DNA binding factor 1, and Wiskott–Aldrich syndrome-like (Wasl) [131] (Figure 1 and Table 3).

The **miR-181** family is an important regulator of T cell development, proliferation, and activation [133]. MiR-181c reduces the SCAR/WAVE actin-nucleating complex unit (BRK1) protein expression level. BRK1 is a member of the WAVE protein family. The miR-181-BRK1 axis is required for actin polymerization and, thus, for lamellipodia formation in T lymphocytes. Moreover, overexpression of miR-181c leads to severe impairment of actin polymerization in response to stimulation and a marked reduction in the amount of F-actin at the T cell-B cell contact site, proving its importance in immunological synapse formation [133].

The homeostasis of mast cells of the innate immune system is also modulated by miRNAs. Monticelli and coworkers showed that **miR-221** influences the effector functions and actin cytoskeleton of mast cells, but does not affect their differentiation. They proposed a model in which miR-221 has a dual function. In resting cells, miR-221 maintains normal mast cell homeostasis via cell cycle regulation and cytoskeleton targeting of Cyclin-dependent kinase inhibitor 1B (p27^Kip1^), CD25, and actin. In contrast, upon mast cell stimulation miR-221 influences the extent of degranulation and cytokine production in response to IgE-antigen complexes. Overexpression of miR-221 resulted in a thicker F-actin ring formation compared to control cells. In contrast, the depletion of miR-221 reduced the amount of F-actin [134]. The major target of miR-221 is p27^Kip1^, which is not only a cell cycle inhibitor but is also important in cell motility, suggesting its synergistic effect on the cytoskeletal changes in mast cells (Figure 1 and Table 3).

### 5.4. Actin Regulation in Podocyte Biology

Podocytes cover the external surface of the glomerular capillary by a sophisticated web of primary and secondary ramifications. The maintenance of the dynamic actin–myosin cytoskeletal architecture is crucial in the biology of podocytes since not only the shape and function, but the foot process effacement and microvillus transformation, are determined by alterations in actin [138]. Brain-derived neurotrophic factor (BDNF) is a pleiotropic neurotrophin found to bind to the tropomyosin-related kinase B (TrkB) in podocytes. BDNF downregulates **miR-134**, directly augmenting LIMK1 translation. Upregulated **miR-132** reduces p250GAP translation, resulting in an attenuation of the blockade of Rac1. In turn, miR-132 increases LIMK1 expression and phosphorylation, which affects cofilin activity and favors actin polymerization [139].

### 5.5. MiRNA Regulation of the Actin Cytoskeleton in Osteoblast Differentiation

Bone morphogenic protein II (BMP-II) is a signaling molecule of the TGF- β superfamily that regulates the actin cytoskeleton by acting on miRNAs. BMP plays a crucial role in bone formation and development [141] by binding to BMP receptors (BMPR) followed by activation of SMAD proteins or MAPKs [143]. **MiR-1187** is a robustly downregulated miRNA in medicarpin-stimulated osteogenesis. Overexpression of miR-1187 results in decreased actin polymerization, cortical protrusion formation, and osteoblast mineralization. MiR-1187 is a repressor of BMPRII and the Cdc42 guanine nucleotide exchange factor 9 (ARhGEF-9) expression, which is involved in the activation of Cdc42, leading to actin cytoskeleton rearrangement [142]. The suggested mechanism for the negative regulation of osteogenic differentiation is that miR-1187 represses BMPR-II and ArhGEF-9, thereby suppressing the non-SMAD/Cdc42 signaling pathway leading to the inhibition of p21- activated kinase (PAK1) and LIMK phosphorylation and the activation of cofilin, thus inhibiting actin polymerization and cytoskeletal rearrangement [140].

### 5.6. MiRNAs as Key Regulators in Actin Reorganization in on Cilia Assembly

Another process, in which actin dynamics is highly controlled, is cilia assembly. Cilia assembly is crucial to several biological processes, including fluid flow, cell movement, and sensory perception in organisms ranging from protozoa to mammals [187]. Cilia are microtubule-based, micrometer-scale, whip-like organelles that can form motile cilium on multiciliated cells (MCCs) on the luminal surface of airways, cerebral ventricles, oviducts, and the efferent ducts of testis. Multiciliogenesis is characterized by several steps. First, MCC precursors exit from the cell cycle, followed by multiplication of centrioles, then the apical actin cytoskeleton is reorganized into a cortical meshwork of actin, and, finally, the synthesized centrioles migrate towards the apical pole of the cell, where they anchor to the actin meshwork and mature into basal bodies from which a cilium can be elongated [147]. **MiR-34/449** miRNAs are key regulators of vertebrate multiciliogenesis that directly repress the expression of multiple proteins. MiR-34/449 downregulates the expression of several cell cycle-regulated genes and members of the Notch pathway, which enhance miR-449 expression and facilitate entry into MCC differentiation. On the other hand, micro-RNA34/449 appears to be a negative regulator of CP110, which is involved in basal body maturation as a protein-blocking ciliation [148]. Lastly, miR-34/449 controls the apical actin meshwork formation by targeting components of the small GTPase pathways directly repressing the small GTPase R-Ras during multiciliogenesis [146].

Another candidate in the control of multiciliogenesis is **miR-129-3p,** which represses multiple CP110 and actin regulators critical for cilia formation. Upon overexpression of miR-129-3p, RPE1 cells exhibited a disorganized actin cytoskeleton with amorphous aggregates and speckles compared to control cells with parallel arrays of bundled filamentous actin or stress fibers. MiR-129-3p facilitates basal body formation by repressing CP110 expression. As a parallel function, miR-129-3p inhibits ciliogenesis and axoneme growth by hampering F-actin formation via targeting and downregulating gene expression of ARP2 (core subunit of ARP2/3 complex), TOCA1 (Cdc42 effector), ABLIM1, and ABLIM3 (homologous F-actin binding proteins) [144,145]. This further establishes miR-129-3p as a central player in multiciliogenesis, acting at several distinct levels of this complex physiological process (Figure 1 and Table 3).

### 5.7. Signaling Pathways of miRNAs-Regulated Cancer Formation

Regulation of the actin cytoskeleton is involved in cancer cell migration, tumor invasion, and metastasis [170]. MiRNA dysregulation is implicated in the development and progression of nearly all tumor types and some miRNAs can function as tumor suppressors (Figure 1 and Table 3).

**MiR-124** serves as a tumor suppressor in various cancers, including breast and prostate cancers [164,173]. MiR-124 is also involved in the development of liver cancer; miR-124 suppresses aggressive hepatocellular carcinoma by repressing Rho-associated kinase 2 and the zeste polycomb repressive complex 2 subunit [172]. A systematic analysis of the molecular mechanism of miR-124 revealed that elements of the ROCK signaling pathway are regulated by the overexpression of miR-124 in hepatocarcinoma cells. In addition, miR-124 regulates a total of 17 genes, which are significantly associated with the actin cytoskeleton pathway, including talin, cofilin 2, Wiskott–Aldrich Syndrom (WAS) protein family member 1 (WASF1), and CFL2, an intracellular actin-modulating protein that regulates actin-filament dynamics [168]. The overexpression of miR-124 represses cell migration and invasion of intrahepatic cholangiocarcinoma cells [171]. An et al. showed that miR-124 is downregulated in malignant glioma cells and its target is the ROCK1 gene. Overexpression of miR-124 reduced ROCK1 expression and suppressed glioma cell invasion by affecting actin cytoskeleton rearrangement and reducing cell surface ruffles [65].

Another candidate in the regulation of the actin cytoskeleton is **miR-1181,** which is downregulated in pancreatic cancer cell lines compared to normal ductal epithelial cells. The overexpression of miR-1181 inhibited the migration, invasion, and proliferation of PANC-1 pancreatic cancer cells and resulted in decreased expression of F-actin and β-tubulin. Furthermore, miR-1181 inhibits the expression of signal transducer and activator of transcription 3 (STAT3) [127].

The development of thyroid carcinoma is also regulated by miRNAs. Has-miR-196a-2 is overexpressed in thyroid carcinoma according to a bioinformatics analysis. A Gene Sets Enrichment analysis suggested that **has-miR-196a-2** is enriched not only in the adherens junction, but in the actin cytoskeleton as well [160], and that has-miR-196a-2 expression is closely related to the invasion and migration of thyroid carcinoma cells due to the upregulation of the WNT pathway [151].

Rho superfamily signaling is controlled at the level of expression of downstream effectors of small GTPases, such as RhoA or C. Both the hyperactivation of ROCK and the downregulation of its upstream regulators have been described in numerous cancer types. ROCK is targeted by **miR-138** [156] and **miR-139** in tongue squamous cell carcinoma (TSCC) and hepatocellular carcinoma (HCC), respectively. Reduced miR-138 levels correlate with increased expression of RhoC and ROCK2, which results in altered, elongated cell morphology, enhanced cell stress fiber formation, and accelerated cell migration and invasion in TSCC [156]. MiR-139 interacts with the 3’ untranslated region of ROCK2 and reduces its expression in HCC cells. Levels of miR-139 inversely correlate with ROCK2 protein in human HCC samples. The expression of miR-139 is reduced in human metastatic HCC samples and correlates with prognosis [169].

Ultimately, signaling by many Rho GTPases via numerous downstream actin-regulating proteins control the dynamics of actin polymerization in cancer. **MiR-31** [166] and **miR-200** [165] are capable of suppressing the WASP-family protein member 3 (WAVE3). The expression of WAVE3, an actin cytoskeleton remodeling and metastasis promoter protein, showed a clear inverse correlation with miR-31 and miR-200 in invasive versus non-invasive cancer cells [165].

The downregulation of several miRNAs modulates cytoskeletal dynamics in breast cancer. These miRNAs primarily affect the upstream regulators of the actin cytoskeleton and myosin modulating cell spreading, the formation of lamellipodia, and cell invasion. The first miRNA-target mRNA pair to be verified in vivo was ***let-7* miRNA,** which negatively regulates target gene expression by either mRNA cleavage or translational repression. Let-7 is the most frequently and significantly associated miRNA with clinical outcomes in patients with cancer [162]. Let-7b is reduced in patients with metastatic breast cancer correlating with cell migration and invasion by affecting focal adhesion and stress fiber stabilization. Let-7b directly represses multiple genes involved in the actin cytoskeleton pathway, including PAK, radixin (RDX), integrin beta-8 (ITGB8), and diaphanous-related formin 2 (DIAPH2). However, the expression of actin remains the same, suggesting that the actin cytoskeleton modulation occurs through the downregulation of its upstream effectors [155].

Another miRNA that regulates the actin cytoskeleton through the regulation of PAK in breast cancer is **miR-23b**. Clinically, low miR-23b correlates with the development of metastases in breast cancer. Expression of miR-23b increases F-actin size and cell spreading as well as the formation of lamellipodia. MiR-23b inhibits a large number of cytoskeletal genes, including LIMK2, CFL2, and PIK3R3, by direct interaction with their 3′UTRs. In contrast, miR-23b downregulates PAK2 by an indirect mechanism and decreases the phosphorylation of myosin light chain (MLC) II without modulating its gene expression [163]. Liu et al. identified **miR-30c** as a potential regulator of the actin cytoskeleton in breast cancer. MiR-30c functions as a tumor suppressor acting directly on twinfilin 1 actin-binding protein and vimentin. Interleukin-11 is a downstream target of twinfilin 1, which regulates STAT3 phosphorylation, causing invasion of cancer cells [149].

A major player in the progression of breast cancer is transforming growth factor- β (TGF- β), which promotes epithelial-to-mesenchymal transition, migration, invasion, and metastasis [161]. TGF- β silences the expression of **miR-584**, resulting in enhanced protein phosphatase and actin regulator 1 (PHACTR1) expression, leading to actin rearrangement and breast cancer cell migration [150].

One of the most predominant miRNAs regulating the formation of stress fiber formation through diverse signaling pathways is **miR-200**. The expression of miR-200 is often downregulated in breast metastases and the reduced expression is associated with poor prognosis in breast cancer [152]. MiR-200c downregulates the transcriptional repressors, ZEB1 and SIB/Zeb2 [153], resulting in the upregulation of E-cadherin and the inhibition of cell motility in cancer cells [159]. However, miR-200c also acts on actin remodeling by directly targeting moesin, fibronectin 1 [154], formin homology 2 domain containing 1 (FHOD1), and Mg^2+/^Mn^2+^-dependent protein phosphatase 1F (PPM1F) [157]. FHOD1 is an actin-nucleating protein which induces the formation of actin filaments by activating serum response factor (SRF), a key regulator of the actin cytoskeleton and contractile processes, resulting in increased MLC expression. Phosphorylation of MLC also contributes to stress fiber formation through cross-linking actin filaments. FHOD1-stimulated actin nucleation also contributes to stress fiber formation, while PPM1F increases the phosphorylation level of MLC independent of MLC expression [157]. PPM1F modulates not only stress fiber formation but also modulates MLC via inhibition of PAK, provoking increased phosphorylation of MYPT of myosin phosphatase at the inhibitory site [158]. Apart from the previously described targets of miR-200c, Brabletz et al. identified the tyrosine kinase substrate with five SH3 domains (TKS5) and MYLK as further direct targets. Expression of TKS5 and MYLK correlates inversely with miR-200c in cancer cells and breast cancer patients. Both TKS5 and MYLK are necessary for invasion and invadopodia formation [167].

### 5.8. MiRNAs in Actin Cytoskeleton Regulation: Leukocytes and Lymphocytes in Action

Tuberculosis is a major cause of death worldwide. *Mycobacterium tuberculosis*, the causative agent in tuberculosis, has evolved an ingenious strategy to survive inside of the host cell. *Mycobacterium tuberculosis* manipulates the early steps of interaction with macrophages (Mφ) to avoid the activation of microbiocidal mechanisms [175]. **MiRNA-142-3p** is a key candidate in the regulation of actin dynamics required for phagocytosis. Expression of miR-142-3p is induced in murine Mφ upon mycobacterial infection, accompanied by a temporal inhibition of N-Wasp (WASL) expression. WASL is involved in the regulation of actin dynamics during the invasion of host cells by pathogens. MiR-142-3p induces a significant decrease in intracellular mycobacterial intake of Mφ [174]. In addition to miRNA-142-3p, other miRNAs, including miR-377, miR-32, miR-410, miR-19b, and let-7f may bind to the 3+-UTR sequence of Was1 upon mycobacterial infection [174].

Epidermal wound healing also requires the activation of an orchestra of miRNAs. **MiR-142** family members are indispensable for protection against *Staphylococcus aureus* infection and clearance at wound sites. Healing of *S. aureus*-infected skin wounds was significantly delayed in miR-142-depleted mice compared with WT mice. Both chemotactic and phagocytic behavior of neutrophils is impaired, including disturbed cell polarity, increased cell motility, altered phagocytosis, and enhanced F-actin assembly. MiR-142 regulates actin cytoskeleton dynamics in neutrophils by downregulating small GTPase translation. MiR-142-3p and miR-142-5p interact with Rac1 and RhoA mRNA 3′-UTRs, downregulating actin polymerization and lamellipodia formation and actomyosin assembly, respectively [178].

MiR-142-3p also regulates the actin cytoskeleton of CD4+ T cells via the previously mentioned mechanism targeting Rac1 and ROCK2 in atherosclerosis. In this chronic immune-inflammatory disease, inflammatory vascular cells release chemokines, such as CXCL12, to attract and activate lymphocytes [176]. CXCL12 inhibits the expression of miR-142-3p in CD4+ T cells, causing the elevated expression of its target genes, RAC1 and ROCK2. Increased RAC1 promotes the assembly of lamellipodia, helping the T cells to protrude toward chemokine signals. On the other hand, ROCK2 activates the actomyosin complex contraction and facilitates the motive power of CD4+ T cells for migration [177].

## 6. Conclusions

MiRNAs play crucial roles in numerous biological processes in various tissues through the regulation of the actomyosin complex. The majority of miRNAs are conserved across multiple species, indicating the importance of these molecules. Many publications in the last decade demonstrate altered miRNA expression in human diseases and selective modulation of miRNAs through antisense inhibition or replacement could significantly affect disease development. In fact, miRNAs are differentially expressed in many diseases producing miRNA signatures, such as let-7 in various cancer types, miR-145 in vascular diseases, and miR-208 in cardiomyocyte hypertrophy [188]. Thus, miRNAs may be valuable drug targets and biomarkers. As master regulators of the human genome, miRNAs are ideal candidates for therapeutic applications. Moreover, recent technological advances, combined with the ease of administration and highly efficient tissue uptake of miRNA, without the need for developing new formulations, give miRNA therapeutics an extra edge.

Several new strategies have been developed in miRNA therapy and they bear great potential. MiRNA-based therapies act via highly specific binding of oligomers and antagomirs to the complementary mRNA target sites, but this technology still needs to overcome multiple challenges [189]. First, miRNAs might have numerous targets and up- or downregulation of a pathogenic or beneficial miRNA can affect an entire gene network or signaling pathway. Moreover, the tissue-specific regulation of miRNA is another challenge, which can potentially be ameliorated by the addition of targeting components to enhance tissue specificity [190]. Third, to find the most suitable, efficient, and specific delivery system in miRNA replacement therapy is a challenge. The efficacy of miRNA-based therapy is greatly decreased by unsuitable vector size, which may be toxic or inactivated by nonspecific serum proteins [191].

One potential application that targets the miRNAs regulating actin and myosin cytoskeleton is the modulation of vascular phenotypes and the use of differential expression markers, such as smooth muscle α-actin, smooth muscle 22α, and smooth muscle myosin heavy chain. Moreover, targeting miRNAs could be utilized in cardiovascular diseases, such as post-angioplasty restenosis, atherosclerosis, and hypertension, where the pathological phenotype switching could be reversed [192]. Another muscle-related application is the ability to manipulate the processes associated with exercise adaptation, including cardiac and skeletal muscle hypertrophies or angiogenesis, by targeting myosin isoforms to induce isotype switches that modulate the contractile and metabolic features of the muscle.

## Figures and Tables

**Figure 1 cells-09-01649-f001:**
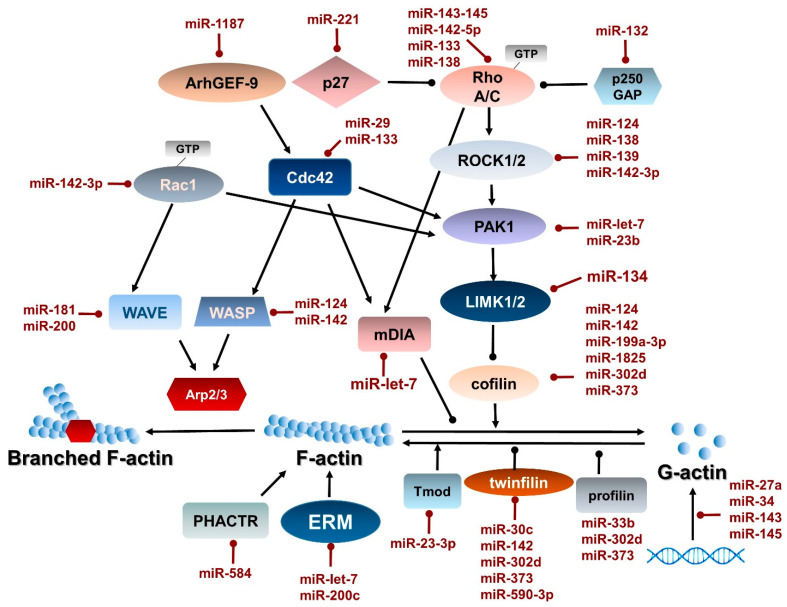
MicroRNAs (miRNAs) involved in the regulation of signaling pathways modulating the actin cytoskeleton. Globular actin (G-actin) forms filamentous actin (F-actin) by the direct action of cofilin, twinfilin, profilin, and tropomodulin (Tmod). F-actin forms branched F-actin with the help of Arp2/3. These processes are modulated by upstream signaling pathways also regulated by microRNAs (miR). Black arrows and round-headed lines represent activation and inhibition, respectively. Lines show the action of miRNAs (red) and proteins regulated by miRNAs (black).

**Figure 2 cells-09-01649-f002:**
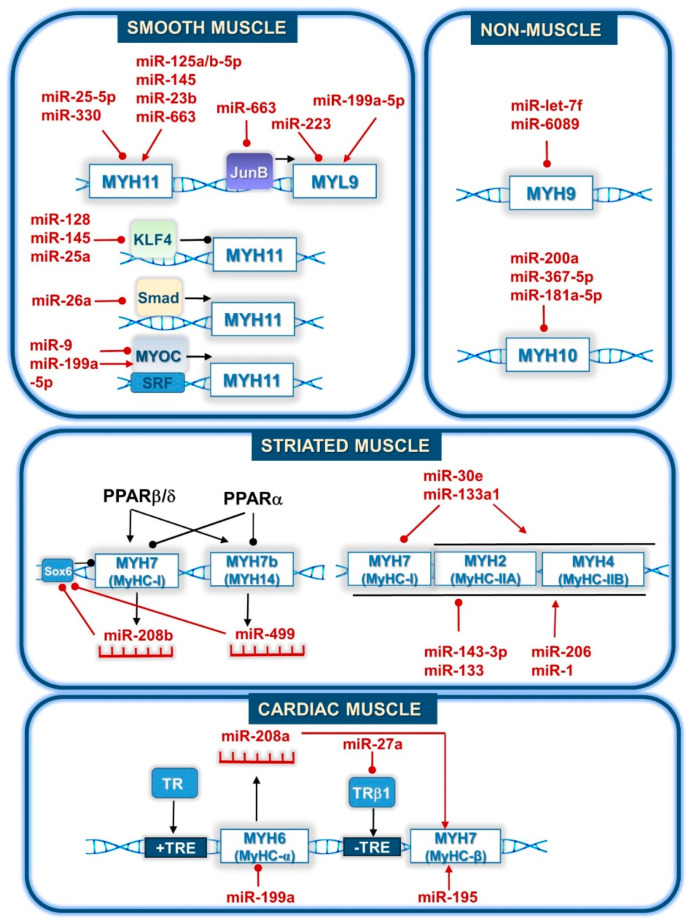
Regulation of myosin isoforms by microRNAs. In smooth muscle, MYH9 is either regulated directly by microRNAs or through the regulation of their specific transcription factors, such as Krüppel-like factor 4 (KLF4), SMAD, and myocardin (Myoc). In striated muscle, PPARβ/δ and PPARα exert contrary effects on MYH7 and MYH7b genes. MiR208b and miR-499 are encoded by the MYH7 and MYH7b genes, respectively, and they affect the same mRNAs to downregulate Sox6. In cardiac muscle, the transcription of MYH7 is activated by the product of MYH6, namely miR-208a. This process is also promoted by miR-195 and by the inhibitory action of miR-27a on the thyroid receptor beta 1(THRβ1) that binds to an inhibitory thyroid response element (-TRE). MicroRNAs (miR) are labeled in red with their numbers. Black arrows and round-headed lines represent activation and inhibition, respectively. Lines show the action of miRNAs (red) and proteins regulated by miRNAs (black). Myosin genes are labeled in the white boxes as follows: MYH10 (myosin heavy chain 10); MYH11 (myosin heavy chain 11); MYH7b (myosin heavy chain 7b); MYH14 (myosin heavy chain 14); MYH9 (myosin heavy chain 9); MyHC (myosin heavy chain); MYH2 (MyHC-IIA, myosin heavy chain-IIA); MYH4 (MyHC-IIB, myosin heavy chain-IIB); MYH7 (MyHC-slow(I), myosin heavy chain slow type I); MYH6 (MyHC-α, myosin heavy chain alpha); MYH7 (MyHC-β, myosin heavy chain beta); MYL9 (myosin light chain 9).

**Table 1 cells-09-01649-t001:** Regulation of actin genes by miRNAs.

miRNA	Physiology	Regulation	Regulated Transcript	Citation
miR-let-7g	smooth muscle contraction	downregulation	α-SMA	[67]
miR-19a	normal smooth muscle contractility	upregulation	α-SMA	[68,69,70]
miR-21	smooth muscle contraction	upregulation	α-SMA	[71]
miR-23b	vascular smooth muscle contraction	upregulation	α-SMA	[72]
miR-25a-5p	vascular smooth muscle contraction	upregulation	α-SMA	[73]
miR-26a	vascular smooth muscle contraction	downregulation	α-SMA	[74]
miR-27a	primary vascular smooth muscle cells	downregulation	α-SMA	[75,76]
miR-34/34b-5p	normal cytoskeletal function	upregulation	α-actin	[66]
miR-143/145	vascular smooth muscle	upregulation	α-actin	[77,78]
miR-182	vascular smooth muscle	upregulation	α-SMA	[79]
miR-330	vascular smooth muscle contraction	downregulation	α-SMA	[80]
miR-548-f	normal smooth muscle contractility	upregulation	α-SMA	[81]
miR-663	vascular smooth muscle contraction	upregulation	α-SMA	[82]

**Table 2 cells-09-01649-t002:** Regulation of myosin genes by miRNAs.

miRNA	Physiological Effect	Regulation	Regulated Transcript	Citation
**miR-1**	myoblast differentiation	upregulation	MYH (MyHC)	[84]
**miR-23b**	vascular smooth muscle contraction	upregulation	MYH11	[72]
**miR-26a**	vascular smooth muscle contraction	downregulation	MYH11	[74]
**miR-27a**	heart	upregulation	MYH7 (MyHC-)	[89]
**miR-30e**	skeletal muscle fiber type formation	upregulation	MYH2, MYH1, MYH4 (MyHC-IIA,IIX,IIB)	[86]
**miR-125a-5p**	vascular smooth muscle contraction	upregulation	MYH11	[73]
**miR-133**	myoblast differentiation	downregulation	MYH (MyHC)	[84]
**miR-133a**	skeletal muscle fiber type switch	upregulation	MYH4 (MyHC-IIB)	[87]
**miR-133a1**	skeletal muscle fiber type formation	downregulation	MYH7 (MyHC-I)	[90]
**miR-143-3p**	myoblast differentiation	downregulation	MYH (MyHC)	[85]
**miR-143-5p**	melanin secretion	downregulation	MYO5A (myosin 5A)	[91]
**miR-145**	melanosome transport	downregulation	MYO5A (myosin 5A)	[92]
**miR-195**	cardiac remodeling	upregulation	MYH7 (MyHC-β)	[93]
**miR-199a**	cardiomyocyte contraction	downregulation	MYH6 (MyHC-α)	[94]
**miR-206**	myoblast differentiation	upregulation	MYH (MyHC)	[83]
**miR-208a**	cardiac remodeling	upregulation	MYH7 (MyHC-β)	[95]
**miR-208b**	skeletal muscle fiber type and energy metabolism	upregulation	MYH7 (MyHC-I)	[96]
**miR-330**	vascular smooth muscle contraction	downregulation	MYH11	[80]
**miR-499**	striated muscle	upregulation	MYH7b/MYH14 (MyHC-14)	[97]
**miR-501**	skeletal muscle regeneration	upregulation	MYH3 (MyHC-emb)	[88]
**miR-663**	vascular smooth muscle contraction	upregulation	MYH11 MYL9	[82]

**Table 3 cells-09-01649-t003:** MicroRNAs implicated in the regulation of the actomyosin cytoskeleton.

miRNA	Function/Pathology	Regulation	Regulated Transcript	Citation
miR-1	cardiomyocyte hypertrophy	downregulation	Rho A, CDC42	[179]
miR-let-7	breast cancer	downregulation	PAK1, DIAPH2, RDX,ITGB8	[155]
miR-let-7g	smooth muscle contractility	upregulation	calponin	[67]
miR-19a	smooth muscle contractility	upregulation	SM22α	[68,69]
miR-21	smooth muscle contractility	downregulation	SM22α tropomyosin	[71,182]
miR-23b	breast cancer vascular smooth muscle	downregulation upregulation	PAK2, LIMK2 SM22α	[163]
miR-23b-3p	erythropoiesis	downregulation	E-Tmod41	[136]
miR-26a	vascular smooth muscle contractility	downregulation	SM22α	[74]
miR-30c	breast cancer	downregulation	twinfilin 1, vimentin	[149]
miR-31	tumorigenesis, metastasis formation	downregulation	RhoA, WAVE3	[165,166]
miR-34/449	ciliation	downregulation	R-Ras CP110	[146,148]
miR-124	liver cancer glioblastoma	downregulation	ROCK2, CRL, WARP, cofilin ROCK1	[168] [65]
miR-128	vascular smooth muscle contractility	upregulation	KLF4	[112]
miR-129-3p	multiciliogenesis			[144,145]
miR-132	podocyte formation	downregulation	LIMK1	[139]
miR-133	cardiomyocyte hypertrophy	downregulation	RhoA, CDC42	[179]
miR-133a	airway smooth muscle	upregulation	RhoA, Cdc42	[179]
miR-134	podocyte	upregulation	P250GAP	[139]
miR-138	tongue squamous cell carcinoma	upregulation	Rho C, ROCK2	[156]
miR-139	hepatocellular carcinoma	downregulation	ROCK2	[169]
miR-142	megakaryopoiesis	Needs to be active for normal	Cfl-2, Wasl, twinfilin, integrin α	[131]
miR-142-3p	phagocytosis T cell activation	upregulation downregulation	WASL, Cfl-2 Rac, ROCK2	[174] [178] [177]
miR-142-5p	T cell activation	upregulation	RhoA	[178]
miR-143/145	vascular smooth muscle atherosclerosis glaucoma	upregulation	MRTF-B, actin, Cfl-2, KLF5, MYOCD, ROCK1 Myosin VI, MYLK	[77,78] [183] [184]
miR-181c	T cell activation	downregulation	BRK1	[185]
miR-182	vascular smooth muscle contractility	upregulation	SM22α, calponin	[79]
Has-miR-196a-2	thyroid cancer	upregulation	WNT	[151]
miR-200c	breast cancer tumorigenesis	down	ERM, fibronectin FHOD1, PPM1F TKS5, MYLK WAVE3 ZEB	[154,157,167] [153,165]
miR-221	mast cell homeostasis and stimulation	upregulation	p27^Kip1,^ CD23	[134]
miR-223	vascular smooth muscle contractility	downregulation	RhoB	[78]
miR-330	vascular smooth muscle contractility	downregulation	calbindin, smoothelin, renin	[80]
miR-584	breast cancer	downregulation	PHACTR	[150]
miR-663	vascular smooth muscle contractility	upregulation	SM22α	[82]
miR-1181	pancreatic cancer	downregulation	β-tubulin STAT3	[127]
miR-1187	osteoblast	downregulation	BMPR-II, ArhGEF-9	[140]

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
