# Peer review of "MicroRNA Regulatory Pathways in the Control of the Actin–Myosin Cytoskeleton"

_cells, 2020, doi:10.3390/cells9071649_

Round 1
Reviewer 1 Report
The authors have submitted a review on miRNA-mediated regulation of the actin-myosin cytoskeleton. While the actin-based cytoskeleton is extremely complex and responsible for a plethora of critical cellular functions, relatively little has been done in terms of studying how miRNAs might directly affect these functions. The submitted manuscript is well-written, very thorough, and is a useful and informative compilation of the current state of this subfield. I found the tables and especially the figures to be valuable tools for summarizing current knowledge. This review should be a welcome addition to the literature and will provide a solid basis of information for those interested in this topic.
I have only a few very minor editorial suggestions for the reviewers:
- Page 11, lines 392 – 399: The term “actin-myosin network” is used very often and redundantly in this section, which makes for a bit awkward reading. Please make revisions here.
- Page 15, line 540: This sentence should read, “…a signature pattern of miRNAs is upregulated…”
- Page 19, line 695: I believe this should read, “…elements of the ROCK signaling pathway are regulated by the overexpression…”
Reviewer 2 Report
Review: “MicroRNA regulatory pathways in the control of the actin-myosin cytoskeleton” by Uray et. al 2019
In this manuscript the authors review literature supporting the role of miRNAs in actin-myosin under physiological and pathological conditions. The review summarizes miRNAs important in hematopoiesis, podocyte physiology and osteogenesis. Except for a few points explained below, I think it is worthy of publication.
Some points to consider:
The text would benefit if it was written in a more concise form. The use of unnecessary words and redundant sentences makes it difficult to read, especially when there is so much information. Avoid nominalizations for clarity and simplicity. The following is an example but there are many throughout the manuscript.
Lines 417-418: “The conditional deletion of Dicer in smooth muscle cells induces decreased differentiation and, consequently, decreased contractile function. Similar results were obtained by deleting Dicer in adult mice [89].
Simplify to maybe:
“Dicer deletion in smooth muscle cells decreases differentiation and contractile function in embryo (?) and adult mice [89].”
Avoid using:
“The authors found…”
“Previous studies have shown that/Emerging evidence suggest that” (you will reference the papers so no need to mention it)
“have been found to be”
“recent results...”
“Respectively”
“Widely” (it is overused)
“is a key element” (overused)
Paraphrase and do not use direct quotes from references without “” and cite the correct reference. E.g.
Lines 437-440: “The molecular mechanisms underlying the regulation of SMC differentiation by these miRNAs are diverse, but involve the direct or indirect regulation of transcriptional factors, such as serum response factor (SRF), myocardin (MYOCD), myocardin-related transcription factors (MRTFs), and the Krüppel-like zinc finger family (KLF).” Direct quote from Li et al 2013, not referenced in the text.
Lines 484-485: “PDGF also promotes the synthetic VSMC phenotype and increases smooth muscle cell proliferation and subsequent migration into the neointima layer after artery injury” Direct quote from Li et al 2013.
Lines 487-490: “The overexpression of miR-663 is associated with the increased expression level of the molecular markers of VSMC differentiation and reduced expression level of the transcription factor JunB and myosin light chain 9 (MYL9)” Direct quote from Li et al 2013.
Lines 562-564: “Non-muscle myosin II (NMII) is a major contributor to cellular organization, polarity, and regulation; altered NMII activity contributes to numerous disease pathologies. NMII crosslinks and slides actin filaments past each other, contracting them into actomyosin filament bundles.” Direct quote from Newell-Litwa, et al 2015.
Lines 566-568: “Non-muscle myosin IIA (NMIIA) is the myosin isoform consisting of non-muscle 566 myosin heavy chain IIA (NMHCIIA), encoded by the MYH9 gene, and regulatory and essential light 567 chains that are shared with other NMII isoforms” Direct quote from Newell-Litwa, et al 2015.
Lines 577-579: “Non-muscle myosin IIB (NMIIB) consists of NMHCIIB, encoded by the MYH10 gene, and regulatory and essential light chains that are shared with other NMII isoforms. In migratory cells, NMIIB preferentially localizes to the rear of the cell.”Direct quote from Newell-Litwa, et al 2015
Lines 590-592: “Non-muscle myosin IIC, the most recently discovered non-muscle myosin isoform, consists of NMHIIC and is encoded by MYH14.” Direct quote from Newell-Litwa, et al 2015
References. Authors need to update references and to use original articles and not reviews as much as possible.
Line 401: Reference needed.
Line 197: “More than 650 miRNAs have been reported in human cells” The reference used is very old. This should be updated . In 2019 miRBase V22 had annotated over 1000 and Alles et al 2019 reported over 2000 human miRNA by high/low-throughput analysis. This numbers may have already changed…
Line 198: “…based on a recent estimation, about 3-4% of human genes encode miRNAs [47].” This reference is relatively old. Either update the reference or do not say it is a recent estimation.
Introduction: Expand on which proteins are involved in miRNA processing (Lines 203-215). Dicer is mentioned in line 415 but there is not previous information about its role in miRNA processing, so it is not clear why “The crucial role of miRNAs in smooth muscle function is demonstrated by Dicer knockouts”
Figures. Figures legends should be more descriptive and detailed so that the reader can understand the figure without going back to the text. Symbols and colors should be explained. Sometimes, the text refers to figure 1 but the information in the text does not match the figure and vice versa (e.g. last paragraph of page 12).
Tables. In table 3, the “Function” column lists both function and pathological outcome. Either have two columns of change the headline. The “Regulated gene” should be “Regulated transcript”
Conclusion. Expand in what are the challenges in the use of miRNAs as therapies.
Reviewer 3 Report
The manuscript submitted by Uray et al. describes the important role of microRNAs in the control of the actin myosin cytoskeleton. All in all, it is a very comprehensive manuscript, which describes all known microRNAs that are involved in regulating actin myosin function. The first part describes the regulation of actin and myosin, followed by a list of all miRNAs. The last part discusses the role of miRNAs in pathophysiological processes. The only point of criticism is that this structure often leads to repetitions. The manuscript would benefit from being streamlined. In its current form, it is very tiring to read. Especially the third part should be revised. The importance of miRNAs in cardiovascular diseases and cancer is of great importance. But for example 5.3, the miRNA regulation of non-muscle myosin, could be omitted completely and integrated in other parts of the manuscript. Likewise, the headings in the third part should be uniformly designed in relation to the diseases. The authors very often repeat what is described in the review. This would be enough once. A small note, if already abbreviated, continue to use the abbreviation, especially for microRNA.
